# Cognitive Function in Peri- and Postmenopausal Women: Implications for Considering Iron Supplementation

**DOI:** 10.3390/nu17111762

**Published:** 2025-05-23

**Authors:** Mun Sun Choi, Emily R. Seiger, Laura E. Murray-Kolb

**Affiliations:** Department of Nutrition Science, Purdue University, 700 Mitch Daniels Blvd., Stone Hall Rm. 214, West Lafayette, IN 47907, USA; mchoi@purdue.edu (M.S.C.); eseiger@purdue.edu (E.R.S.)

**Keywords:** iron, iron deficiency (ID), iron deficiency anemia (IDA), perimenopause, menopause transition (MT), perimenopausal menorrhagia (PM), abnormal uterine bleeding (AUB), postmenopause, midlife, women’s health, cognitive function

## Abstract

Menopause is associated with significant hormonal and reproductive changes in women. Evidence documents interindividual differences in the signs and symptoms associated with menopause, including cognitive decline. Hypothesized reasons for the cognitive decline include changes in hormone levels, especially estrogen, but study findings have been inconsistent. Hormone replacement therapies (HRTs) are often recommended to alleviate menopause-related symptoms in both peri- and postmenopausal women. However, the North American Menopause Society does not recommend the use of HRT for the management of cognitive complaints in perimenopausal women due to lack of evidence. Additionally, there are many women for which the use of HRT is contraindicated. As such, it would be helpful to have an alternative method for alleviating symptoms, including declines in cognition, during the menopause transition. Iron supplementation may be a promising candidate as it has been associated with improved cognitive performance in premenopausal women with iron deficiency and iron deficiency anemia. Because many women will experience heavy blood losses during perimenopause, they are at risk of becoming iron deficient and/or anemic. The use of iron supplementation in women with iron deficiency may serve to not only improve iron status but also to alleviate many of the signs and symptoms associated with perimenopause (lethargy, depressed affect, etc.), including cognitive decline. However, evidence to inform treatment protocols is lacking. Well-designed studies of iron supplementation in perimenopausal women are needed in order to understand the potential of such supplementation to alleviate the cognitive decline associated with perimenopause.

## 1. Introduction

According to the World Health Organization (WHO), the number of individuals over the age of 60 exceeded the number of children younger than the age of 5 in 2020. In anticipation of many countries continuing to observe a shift in the population age structure towards older ages, definitions of healthy aging were formalized when the WHO developed the Global Strategy and Action Plan on Aging and Health for 2016–2020 [1]. Following its release, the number of new scientific publications listed in PubMed related to healthy aging increased exponentially, peaking in 2021 when more than 5300 healthy aging articles were published. While definitions of healthy aging vary by field, the Global Strategy and Action Plan on Aging and Health for 2016–2020 defines healthy aging as the “process of developing and maintaining the functional ability that enables well-being in older age” with the emphasis on acknowledging diversity in the trajectory and process of aging and reducing inequity.

In order to maintain functional ability with aging, the WHO stresses the importance of maintaining interactions between intrinsic capacity, the combination of an individual’s physical and mental capacity, and environmental characteristics, such as social networks and services provided by communities [2]. In aging women, menopause, the point marking the end of women’s reproductive years, may play a role in disrupting these interactions [3]. Menopause is clinically defined as amenorrhea, or absence of menstrual bleeding, lasting for at least 12 months. It is a normal physiological process due to loss of ovarian follicular function and decline in estrogen levels [4]. As women reach menopause, they are affected by varying degrees of physical, mental, and/or emotional changes [5]. There are many signs and symptoms that are associated with menopause, including vasomotor (i.e., hot flashes, night sweats, insomnia), genitourinary (i.e., abnormal uterine bleeding, vaginal dryness), cardiovascular (i.e., hypertension), and psychogenic symptoms (i.e., anxiety, depression), which may persist and/or worsen throughout menopause and postmenopause [6].

It is estimated that approximately 47 million women become menopausal each year, and the world population of menopausal and postmenopausal women will increase to 1.2 billion by 2030 [7]. In addition, women in high-income countries with access to advanced medical care are expected to live approximately 40% of their lives postmenopausal [8]. Signs and symptoms associated with menopause are shown to have a significant impact on work outcomes, and it is estimated that USD 1.8 billion is lost annually in the United States (US) based on missed workdays related to menopause [9]. Despite its prominent impact on a large portion of the world population as well as much of the world’s economy, less than one percent of pre-clinical trials related to the biology of aging include the modeling of menopausal phenotypes observed in humans, with even fewer focusing on perimenopause [3].

Perimenopause refers to the menopause transition—the time of menstrual irregularity and significant hormonal and reproductive changes leading up to the beginning of menopause [10,11]. Perimenopause typically begins in the fifth decade of life, but there are differences in the onset, duration, and severity of signs and symptoms, with many of the symptoms remaining ill-defined. Thus, the diagnosis is made retrospectively, based on the patient’s age and complaints of signs and symptoms, which contributes to most women facing these physical, emotional, and cognitive changes unprepared [6]. Yet, menopause is an inevitable phase in women’s lives that has strong potential to influence quality of life (QOL) and family dynamics [12].

Cognition is an umbrella term that encompasses various aspects of intellectual functions and processes, including attention, memory, decision-making, planning, comprehension, and reasoning [13]. Cognitive decline is often observed with aging and considered part of normal physiology [14]. In addition, various factors including hormonal fluctuations may exert a great influence on women’s brain functions as they go through different stages of menopause, leading to varying degrees of depressive symptoms, insomnia, and cognitive dysfunction [15,16,17,18]. Thus, the early management of subjective cognitive decline is recommended to maintain QOL and to mitigate the risks of neurodegenerative diseases, especially Alzheimer’s disease, in later life [19,20].

The WHO has identified iron deficiency (ID) as a major public health concern due to its high prevalence, especially in women [21]. If left untreated, ID can lead to anemia. The worldwide prevalence of anemia in women over the age of 60 is approximately 14%, with iron deficiency anemia (IDA) being the most common etiology [22]. Iron plays a crucial role in brain function through oxygen transportation, myelination, neurotransmitter synthesis and regulation, synaptogenesis, and energy expenditure [23,24,25,26]. Thus, the maintenance of a normal physiological brain function requires proper iron homeostasis. Aging is shown to cause disruptions in iron homeostasis as well as redistribution of iron within the brain due to inflammation and changes in the permeability of the blood–brain barrier, which may lead to neuroinflammation and neurodegeneration [24,27,28,29].

One of the many adverse outcomes associated with ID and IDA in adults is diminished cognitive function [30,31]. In young women of reproductive age, ID and IDA negatively affect attention, memory, learning, and behavior [23,32,33,34], and several intervention studies have shown improved cognitive function in those supplemented with iron, suggesting the reversibility of cognitive changes associated with ID and IDA in adulthood. Even in the absence of anemia, ID is associated with a significant decline in cognitive function in geriatric patients [35]. However, studies have not elucidated which specific aspects of cognitive function are affected by ID in the older population, nor have they examined whether iron supplementation is effective at improving cognitive function in this population.

One of the menstrual irregularities that can occur during the menopausal period is perimenopausal menorrhagia (PM). PM, or perimenopausal abnormal uterine bleeding, is defined as abnormally heavy bleeding during the menopause transition. Such heavy bleeding can place a woman at risk of the development or worsening of ID and IDA. PM is estimated to affect at least 25% of perimenopausal women and, in the US, it is estimated that more than 90% of women will experience at least one episode of PM and 78% of those will experience at least three episodes. In addition, abnormal uterine bleeding accounts for 14% of all US hospitalizations among women aged 45–54 years [36]. Women experiencing PM frequently complain of fatigue and impaired work performance, mimicking complaints raised by women with ID and IDA [37,38]. Despite this knowledge, the prevalence of ID and IDA in perimenopausal women and their functional consequences have received little attention.

Based on the high prevalence of PM and existing evidence regarding the association between iron status and cognitive function, assessing and managing women’s iron statuses may be a relatively simple and cost-effective way to mitigate cognitive decline in perimenopausal women. However, before specific recommendations can be made, intervention studies are needed in this population to better understand the relationship between women’s cognitive functions and iron statuses during perimenopause. The purpose of this article is to review the current evidence regarding the impact of different menopausal stages on women’s iron statuses and cognitive functions in order to uncover knowledge gaps where more evidence is needed so that interventions to appropriately manage the problems associated with ID/IDA in perimenopausal women can be developed.

## 2. Literature Search

For the purpose of this narrative review, a literature search was conducted in PubMed, Embase, MEDLINE, Scopus, Web of Science, and CINAHL, where the search terms used include iron, perimenopausal menorrhagia, abnormal uterine bleeding, perimenopause, postmenopause, menopause, cognition, learning, executive function, attention, verbal, and memory. Inclusion criteria were articles investigating healthy human subjects who are classified as late pre-, peri-, and postmenopausal and not using hormone replacement therapy (HRT). A total of 1363 studies were initially identified, but 351 were duplicates. Thus, 1012 studies were screened. During screening, 611 studies were identified to be irrelevant due to the (1) inclusion of patients with chronic illness such as cancer, renal disease, and heart disease, (2) research question, intervention, and/or outcome not relevant for the scope of this review, (3) inclusion of patients with cognitive dysfunction and/or mental disorders such as schizophrenia and autism, (4) no objective measurement of cognitive function, (5) no or unclear inclusion of peri- and/or postmenopausal women, (6) use of psychoactive drugs as the intervention and/or inclusion of psychoactive drug users, (7) article type/study design not relevant, (8) use of HRT as the intervention, (9) use of animal models, (10) inclusion of women undergoing/who underwent surgical menopause, and (11) published in a language other than English. Within 401 studies that were assessed for eligibility, 357 were excluded due to there being (1) no measurement of a relevant outcome of interest, (2) intervention not aligned with the review question such as the use of HRT, (3) article type/study design not relevant, (4) use of animal models, (5) inclusion of HRT users, (6) no clear differentiation of menopause stages, (7) inclusion of women with surgically induced menopause, (8) inclusion of men within analyses, (9) inclusion of women with metabolic syndrome and/or chronic disease, (10) full text not available, (11) published in a language other than English, (12) article retracted, and (13) inclusion of women on psychoactive drugs. In total, 44 studies are discussed in this review article (Figure 1).

## 3. Iron in the Peri- and Postmenopausal Periods

Menopause not only marks a reproductive transitional stage but also a neurological transitional stage, which is associated with symptoms such as insomnia, mood changes, and cognitive decline [19,39,40]. Approximately 44–62% of women experience subjective cognitive decline during the menopause transition, often persisting into postmenopause [41,42,43]. Perimenopausal women have an option to receive HRT to manage signs and symptoms related to the beginning of menopause. Limited evidence suggests that early initiation and continued use of HRT after perimenopause may be associated with better verbal recognition and enhanced hippocampal function later in life [44]. However, the North American Menopause Society does not recommend the use of HRT during perimenopause for the purpose of managing cognitive complaints due to the lack of evidence [45]. In addition, the effectiveness of HRT is not guaranteed, and it may be contraindicated due to a family or medical history such as cardiovascular disease [46,47]. Therefore, identifying other possible interventions, such as iron supplementation, for perimenopausal women who are experiencing cognitive changes could be useful. Given the link between iron status and cognitive function in premenopausal women, iron supplements may be a prospective treatment to alleviate cognitive decline in perimenopausal women [23,30,31,32] but current evidence to support this is sparse, especially in those with PM [48].

Valentina et al. [49] investigated the relationship between ferritin concentrations and cognition (composite score of episodic memory, phonemic/semantic fluency, working memory, and mental flexibility) in perimenopausal women. Surprisingly, the authors found some support for a role of lower ferritin concentrations in overall better cognitive performance; however, these results should be interpreted with caution (Table 1) [49]. First, iron status was assessed only at the beginning of the study while cognitive performance was assessed only at follow-up, which was 13 years later. Thus, there was no control for concurrent iron status when cognitive performance was assessed. Due to iron status changing throughout the life course depending on food and/or supplement intake, changes in absorption, and other physiological changes, it is difficult to definitively link ferritin status from 13 years prior to the measured cognitive performance. Second, ferritin is a well-known acute-phase protein, but no markers of inflammation were measured in the study. Therefore, the association between ferritin and cognition may have been confounded by inflammation.

On the other hand, a recent study by Barnett et al. investigated the relationship between cognitive performance and systemic and brain iron levels in menopausal women with iron sufficiency and ID without anemia. A higher serum ferritin percentile was associated with higher accuracy, higher discriminability, and shorter reaction times in all cognitive tasks assessed (Table 1). In addition, higher values of iron biomarkers that are associated with oxygen transport such as hemoglobin, red blood cell count, hematocrit, mean corpuscular volume, mean corpuscular hemoglobin, and mean corpuscular hemoglobin concentration were related to a better performance in cognitive tasks. The positive relationship between cognitive performance and systemic iron levels observed in menopausal women in the study by Barnett et al. highlights the clinical significance of iron and need for additional studies to confirm these findings, especially in perimenopausal women [50]. Specifically, studies that are adequately powered and, preferably, randomized controlled trials investigating the impact of iron supplementation on iron status and cognitive function in perimenopausal women with ID and IDA are needed.

Postmenopausal women’s iron statuses typically improve over time as iron is no longer lost through menstruation [51]. Additionally, during menopause, an inverse relation between decreasing estrogen and increasing iron levels, specifically serum ferritin concentrations, has been reported [52,53,54]. While there are reasonable concerns regarding a link between increased iron storage in postmenopausal women and an increased risk of cardiometabolic diseases, cancer, and neurodegenerative disorders, the prevalence of elevated iron stores in postmenopausal women is low [53,55,56]. On the other hand, the prescription of HRT, which is associated with lower iron stores in women, is steadily increasing [46,56]. Additionally, postmenopausal bleeding, which accounts for two-thirds of all gynecologic office visits in postmenopausal women, and increased malabsorption of iron as women age also contribute to IDA in postmenopausal women [57,58]. Thus, while the absence of menstruation is typically accompanied by an improvement in iron status in postmenopausal women, not all postmenopausal women are protected from ID and/or IDA and, therefore, may still be susceptible to the adverse cognitive effects of ID [59]. However, as stated above, specifics regarding cognitive effects of ID in postmenopausal women remain unknown, due to an insufficient number of research studies investigating this association.

In particular, there is a lack of direct evidence linking cognitive function and iron status in peri- and postmenopausal women. Emerging evidence [50] suggests that changes in cognitive function observed in perimenopausal women with ID appear to mimic changes in cognitive function in premenopausal women with ID and IDA [23,32,33,34]. In order to adequately examine iron supplementation as a plausible solution to mitigate cognitive decline in perimenopausal women with ID and IDA in future studies, the impacts of menopause stages, menopausal signs and symptoms, and the duration of reproductive life on cognitive function in peri- and postmenopausal women are further explored in this review.

## 4. Cognitive Function in the Peri- and Postmenopausal Periods

A significant decline of estrogen during menopause has been associated with lower cognitive performance in mid- and later life [60,61,62]. Estrogen is responsible for the modulation of neurogenesis and synaptic plasticity and cognitive processing, as shown in functional magnetic resonance imaging (fMRI) [63]. Thus, a decline in estrogen concentration, especially in the brain, affects neural processes through genomic and non-genomic actions that influence the neuronal number, gene expression, and glucose metabolism [19,64]. In addition, estrogen receptors are widely distributed in various regions of the brain, and changes in the concentration of estrogen in the brain may trigger dysregulation of estrogen signaling, leading to changes in neurological function [19,65,66].

A significant portion of perimenopausal women experience cognitive decline [18]. Cognitive domains that appear to be most affected during perimenopause include working memory, attention, processing speed, and verbal memory [67,68,69,70]. These cognitive changes may be associated with hormonal fluctuations and menopausal signs and symptoms such as insomnia, stress, and pain, ultimately affecting perimenopausal women’s QOL [71,72]. Along with other conditions that emerge during perimenopause such as depression and hot flashes, cognitive decline is associated with an increased risk of neurodegenerative diseases in the future if not intervened with [19,20]. Thus, the management of cognitive function is crucial during this stage of menopause.

The impact of perimenopause on cognitive function has been relatively well investigated, but studies regarding the specific domains affected have yielded mixed results. In longitudinal studies conducted by Maki et al., Greendale et al., and Fuh et al., where different domains of cognition were assessed (Table 2), perimenopausal women demonstrated varying degrees of cognitive decline, specifically in learning [67], verbal fluency [69], verbal memory [68], processing speed [68], and attention/working memory [67]. However, Meyer et al. reported that women in the premenopausal and early perimenopausal phases had improved working memory and perceptual speed performance at yearly follow-ups, compared to their first assessment [73]. Cross-sectional studies by Chalise et al., Chen et al., Coslov et al., and Mathew et al. reported that perimenopausal women experienced more frequent and/or severe symptoms associated with poor memory and brain fog than pre- and postmenopausal women [70,74,75,76]. In contrast, Zhang et al. reported that postmenopausal women experienced more severe and frequent “cognitive symptoms”, specifically hypomnesia and lack of concentration, than perimenopausal women in China [77] (Table 2).

Neuroimaging technologies have also been utilized to investigate the association between perimenopause and women’s cognitive performance [78,79,80]. Zhang et al. and He et al. used resting-state fMRI and calculated regional homogeneity values, which reflect the synchronization of neuronal activity in the local brain region [78,80,81]. Both studies found that perimenopausal women had altered patterns of regional homogeneity compared to pre- [78] and postmenopausal women [80], which may be correlated with clinical measures of cognitive function (Table 2) [78]. However, these results should be interpreted with caution as these brain activities were not measured during cognitive task performance. In addition, He et al. compared the cognitive performance of perimenopausal women to that of premenopausal women [78], while Zhang et al. compared the cognitive performance of perimenopausal women to that of postmenopausal women [80]. Lastly, a cross-sectional study reported that perimenopausal women had significantly smaller subcortical volumes in the left and right amygdala compared to premenopausal women, and such structural changes in the bilateral amygdala were related to lower working memory accuracy and a longer executive reaction time. However, the study was conducted using magnetic resonance imaging scans, which may not necessarily reflect what fMRI scans may portray [79]. In order to fully understand the degree of change in cognitive performance observed during perimenopause, it is imperative to compare cognitive performance during pre-, peri-, and postmenopause, ideally within one study, and such studies are largely lacking.

**Table 2 nutrients-17-01762-t002:** Studies examining the relationship between perimenopause and cognitive function.

Author	Study Design	Sample Size	Menopause Stage Investigated	Menopause Stage Categorization Criteria	Cognition-Related Measure(s)	Findings
Chalise et al. (2022) [74]	Cross- Sectional (Nepal)	180	Perimenopause	Menstrual bleeding patterns/history	Face-to-face interview assessing menopause-related symptoms (physical, sexual, and psychological problems associated with menopause) and perception towards menopause	Anxiety, poor memory, and irritability were the most commonly reported psychological problems in perimenopausal women
Chen et al. (2007) [70]	Cross- Sectional (China)	353	Peri- and postmenopause	Menstrual bleeding patterns/history	Menopause-specific Quality of Life Questionnaire assessing the impact of menopausal symptoms on quality of life	“Experiencing poor memory” was the most frequent symptom reported by peri- and postmenopausal women
Coslov et al. (2021) [75]	Cross- Sectional (US)	1529 (583 being perimenopausal)	Late pre- and perimenopause	STRAW+10	Online surveys assessing menopause-related symptoms and their frequency and degree of bother	Perimenopausal women experienced more frequent symptoms associated with brain fog (i.e., forgetfulness, difficulty with concentrating, and difficulty with decision-making) than premenopausal women
Fuh et al. (2006) [69]	Longitudinal (Taiwan)	495	Pre- and perimenopause	Menstrual bleeding patterns/history	Learning (Rey Auditory-Verbal Learning Test), visual recognition (Continuous Recognition Paradigm of Kimura), verbal fluency (Verbal Fluency Test), executive function (Trail-making Test Parts A and B), and attention (Digit Span Test)	Women who became perimenopausal at follow-up performed better in all of the cognitive test except for the learning domain compared to their baseline, but this may be due to learning effects Improvement in verbal fluency at follow-up was significantly less in women who became perimenopausal compared to women who remained premenopausal
Greendale et al. (2009) [68]	Longitudinal (US)	2362	Pre-, early peri-, late peri-, post-, and postmenopause with current hormone use	SWAN criteria (similar to STRAW)	Verbal memory (East Boston Memory Test), processing speed (Symbol Digit Modalities Test), working memory (Digits Span Backward Test)	The Symbol Digit Modalities Test scores did not improve over time with repeated administration in late perimenopausal women The East Boston Memory Test scores did not increase during early and late perimenopause
He et al. (2021) [78]	Cross- Sectional (China)	57 (25 being perimenopausal)	Pre- and perimenopause	STRAW+10	fMRI ReHo Mini-Mental State Examination assessing cognitive impairment	Perimenopausal women had elevated ReHo values in the right lingual gyrus and diminished ReHo values in the right superior frontal gyrus compared to premenopausal women ReHo values of the right superior frontal gyrus were positively correlated with Mini-Mental State Examination scores
Maki et al. (2021) [67]	Longitudinal (US)	443	Pre-, early peri-, late peri-, and postmenopause	SWAN criteria (similar to STRAW)	Attention/working memory (Letter–number Sequencing Test), executive function (Trail-making Test Part B and Stroop Color-Word Trial), processing speed (Symbol Digit Modalities Test and Stroop Color-WordTrial), memory (Hopkins Verbal Learning Test—Revised), learning (Hopkins Verbal Learning Test—Revised), and fine motor skills (Grooved Pegboard Task)	Overall women showed longitudinal declines in continuous measures of learning, memory, and attention/working memory from pre- to early perimenopause From the pre- to early perimenopause, the odds of impairment in learning, memory, and attention/working memory significantly increased by 60%, 71%, and 41%, respectively
Mathew et al. (2021) [76]	Cross- Sectional (India)	315	Peri- and postmenopause	Menstrual bleeding patterns/history	House-to-house surveys assessing menopause-related symptoms and morbidity profile	Poor memory and diabetes were the most common complaints in perimenopausal women
Meyer et al. (2003) [73]	Longitudinal (US)	868	Pre-, early peri-, late peri-, and postmenopause	Menstrual bleeding patterns/history	Working memory (Digit Span Backward Test) and perceptual speed (Symbol Digit Modality Test)	Working memory and perceptual speed performance scores improved for pre- and early perimenopausal women over time as evidenced by yearly improvement in scores at follow-up
Zhang et al. (2021) [77]	Prospective (China)	4063 (2107 being perimenopausal)	Peri- and postmenopause	STRAW+10	Structured questionnaires assessing menopause-related symptoms (negative mood, cognitive symptoms, sleep disorder, vasomotor symptoms, urogenital symptoms, autonomic nervous disorder, and limb pain/paresthesia)	Postmenopausal women experienced more severe and frequent cognitive symptoms than perimenopausal women
Zhang et al. (2021) [80]	Cross- Sectional (China)	50 (25 being perimenopausal)	Peri- and postmenopause	STRAW+10	Processing speed (Stroop Color-Word Trial) rs-fMRI ReHo MRS Patient Health Questionnaire to screen and assess the severity of depression	Compared to postmenopausal women, the ReHo values in the left lingual gyrus and the right precentral gyrus were significantly elevated while the ReHo values in the left inferior temporal gyrus and bilateral putamen were significantly diminished in perimenopausal women There was a significant positive correlation between reaction time and the ReHo value of the left inferior temporal gyrus in perimenopausal women
Zhang et al. (2021) [79]	Cross- Sectional (China)	99 (45 being perimenopausal)	Pre- and perimenopause	Irregular menstrual cycle length FSH Estradiol	MRI Executive function (Stroop Test) and memory (Two-Back Working Memory Task)	Perimenopausal women had significantly smaller subcortical volumes in the left and right amygdala Perimenopausal women exhibited a longer Stroop Test reaction time and Two-Back Working Memory reaction time than premenopausal women Perimenopausal women had a significantly lower accuracy rate in the Two-Back Working Memory Task premenopausal women Perimenopausal women had significantly higher FSH levels than premenopausal women, which was negatively correlated with working memory accuracy and positively correlated with working memory reaction time The Two-Back Working Memory accuracy rate was positively correlated with the left and right amygdala, while the executive reaction time was negatively correlated with the left and right amygdala

fMRI: Functional Magnetic Resonance Imaging; FSH: Follicle-Stimulating Hormone; MRI: Magnetic Resonance Imaging; ReHo: Regional Homogeneity; rs-fMRI: Resting State-Functional Magnetic Resonance Imaging; STRAW: Stages of Reproductive Aging Workshop; SWAN: Study of Women’s Health Across the Nation; US: United States.

Out of many cognitive domains, evidence suggests that working memory, verbal learning, verbal memory, phonemic verbal fluency, attention, and motor function may be more affected in postmenopausal women than pre- and perimenopausal women [43,82,83]. However, results regarding the association between hormones and changes in these cognitive domains in postmenopausal women are not consistent. Berent-Spillson et al. and Weber et al. found that postmenopausal women performed worse in verbal tasks than women in other stages of menopause [43,82]. Ryan et al. found that higher total and free estradiol levels and a higher ratio of testosterone to estradiol were associated with better semantic memory in postmenopausal women. In addition, Ryan et al. found that lower total testosterone and a lower ratio of testosterone to estradiol were associated with better verbal episodic memory in postmenopausal women [84]. In contrast, Epperson et al., Luetters et al., and Herlitz et al. found no associations between endocrine measures and cognitive function across menopause stages [85,86,87] (Table 3).

Lastly, evidence suggests that structural variability in the brain throughout menopause may explain changes in cognitive performance in postmenopausal women. A recent study by Lissaman et al. reported that advanced age in postmenopausal women was associated with lower spatial context memory and microstructural variability in frontal white matter [88]. Jacobs et al. [89] reported a strong correlation between working memory performance and dorsolateral prefrontal cortex–hippocampus connectivity in postmenopausal women, similar to results published by the same group as well as Zhang et al. [90,91]. Later, Seitz et al. found that postmenopausal women showed a positive correlation between regional volumes of anterior cingulate cortex and regional volumes of dorsolateral prefrontal cortex, hippocampus, and inferior parietal cortex, which are regions associated with memory circuitry [92]. A loss of the ability to decrease the resting-state connectivity of left–right hippocampus during the verbal encoding task in postmenopausal women was reported in a recent study, suggesting that altered resting-state functional connectivity in the default mode network may explain changes in memory performances during postmenopause [93] (Table 3).

**Table 3 nutrients-17-01762-t003:** Studies examining the relationship between postmenopausal status and cognitive function.

Author	Study Design	Sample Size	Menopause Stage Investigated	Menopause Stage Categorization Criteria	Hormones Measured	Cognition-Related Measure(s)	Findings
Berent- Spillson et al. (2012) [82]	Cross- Sectional (US)	67 (32 being postmenopausal)	Pre-, peri-, postmenopause	FSH Menstrual bleeding patterns/history	Estradiol FSH	Executive functioning (Trail-making Test), processing speed (Letter and Pattern Comparison Task), visual memory (Brief Visual Memory Test—Revised), verbal memory (Wechsler Memory Scale—Third Edition) Measure of general intelligence (Shipley Institute of Living Scale) Verbal fluency assessed during the fMRI scanning Sleep quality (self-rated sleep quality assessment)	Postmenopausal women had significantly diminished verbal fluency compared to pre- and perimenopausal women Regions involved in the verbal task (right and left inferior frontal cortex, left prefrontal cortex, and left temporal pole) had significant differences in activation between stages of menopause Estradiol and visual memory retention were positively correlated Increasing estradiol and decreasing FSH were correlated with verbal fluency
Epperson et al. (2013) [87]	Longitudinal (US)	403	Pre-, late pre-, early peri-, late peri-, early postmenopause	Menstrual bleeding patterns/history Number of menstrual periods between assessments Cycle length STRAW	Estradiol FSH DHEA Inhibin B LH	Immediate and delayed verbal recall (Buschke Selective Reminding Test), processing speed (Digit Symbol Substitution Test & Symbol Copy Task) Depressive symptoms (Center for Epidemiologic Studies Depression Scale), anxiety (Zung Self-Rating Anxiety Scale), and stress (Perceived Stress Scale)	Immediate and delayed verbal recall performances declined from pre- to postmenopause When adjusted for covariates, only the association between DHEA and processing speed performance remained significant
Herlitz et al. (2007) [86]	Longitudinal (Sweden)	242 (55 being postmenopausal)	Pre-, peri-, postmenopause	Self-reported stages	Estrogen	Episodic memory *, semantic memory *, verbal fluency *, visuospatial skills (Block Design Task), face recognition *	No differences in cognitive performance were found between pre-, peri-, and postmenopausal women No association was found between cognitive performance and estrogen
Jacobs et al. (2016) [90]	Cross- Sectional (US)	186 (31 being postmenopausal)	Pre-, peri-, postmenopause	STRAW+10	Estradiol FSH	Working Memory N-Back Task and Recognition Memory Task completed during fMRI	Bilateral hippocampal connectivity was significantly greater in postmenopausal women than pre- and perimenopausal women Postmenopausal women in the highest tertile of memory performance had a significantly lower bilateral hippocampal connectivity than those who are in the lowest and middle tertiles
Jacobs et al. (2016) [89]	Cross- Sectional (US)	142 (20 being postmenopausal)	Pre-, peri-, postmenopause	STRAW+10	Estradiol FSH LH Progesterone Testosterone	Working Memory N-Back Task completed during fMRI	In postmenopausal women, the connectivity between DLPFC and hippocampus was enhanced compared to premenopausal women The correlation between working memory performance and DLPFC-hippocampus connectivity was stronger in postmenopausal women than premenopausal women
Lissaman et al. (2024) [88]	Cross- Sectional (Canada)	96 (34 being postmenopausal)	Pre- and postmenopause	STRAW+10	Estradiol FSH	Spatial context memory task during the fMRI scanning (Face-Location Memory Paradigm)	Age and postmenopause status were positively correlated with the mean reaction time, but negatively correlated with mean accuracy during the spatial context memory task Advanced age was associated with lower fractional anisotropy and higher mean diffusivity, especially in the anterior corona radiata and genu of the corpus callosum, independent of menopause stage and sex Lower fractional anisotropy and higher mean diffusivity with advanced age were associated with lower spatial context retrieval accuracy in postmenopausal women
Luetters et al. (2007) [85]	Cross- Sectional (US)	1657 (342 being postmenopausal)	Pre-, early peri-, late peri-, and postmenopause	SWAN criteria (similar to STRAW)	Estradiol FSH	Verbal memory (East Boston Memory Test), processing speed (Symbol Digit Modalities Test), working memory (Digits Span Backward Test)	No association was found between stages of menopause and cognitive performance No association was found between cognitive performance and hormone levels
Ryan et al. (2012) [84]	Longitudinal (Australia)	148	Postmenopause	Does not specify	Estrone Total and free estradiol Total and free testosterone SHBG	Verbal episodic memory (Unrelated Word List Test—Immediate Recall, Unrelated Word List Test—Delayed Recall, California Verbal Learning Test—Immediate Recall, California Verbal Learning Test—Delayed Recall), visual episodic memory (Faces Test—Immediate Recognition, Faces Test—Delayed Recognition), semantic memory (Boston Naming Test, Category Fluency Test), and executive function and visuospatial skills (Trail-making Test Part B, Symbol-Digit Modalities Test, Letter–number Sequencing Test, Judgment of Line Orientation Test, Block Design Test)	Better semantic memory performance was associated with higher total and free estradiol levels and a lower testosterone-to-estradiol ratio Better verbal episodic memory was associated with lower total testosterone and lower testosterone-to-estradiol ratio Lower free testosterone levels were associated with greater two-year improvement in verbal episodic memory A higher testosterone-to-estradiol ratio predicted greater semantic memory improvement
Seitz et al. (2019) [92]	Cross- Sectional (US)	94 (32 being postmenopausal)	Pre-, peri-, and postmenopause	STRAW+10	Estradiol FSH	Verbal fluency (Controlled Oral Association Test), memory (Wechsler Memory Scale—Third Edition), verbal IQ (National Adult Reading Test), and episodic memory (Buschke Selective Reminding Test, Face Name Associative Memory Task) Anxiety (Spielberger State-Trait Anxiety Inventory), mood (Profile of Mood Questionnaire), and sleep (Pittsburgh Sleep Quality Index, Insomnia Severity Index) MRI	Postmenopausal women had significant associations between regional volumes of anterior cingulate cortex and regional volumes of hippocampus, DLPFC, and inferior parietal cortex Better memory performance in postmenopausal women was correlated with the enhanced association strength of hippocampus and anterior cingulate cortex
Spets et al. (2024) [93]	Cross- Sectional (US)	180 (29 being postmenopausal)	Pre-, peri-, and postmenopause	STRAW+10	Estradiol Progesterone Testosterone	Memory (Face Name Associative Memory Task) Verbal Encoding Memory Task completed during fMRI	During the Verbal Encoding Memory Task, a significant positive association between rsDMN and left–right hippocampus connectivity was found in postmenopausal women only The positive association observed in postmenopausal women was driven by those with the lowest tertile of memory performance
Weber et al. (2013) [43]	Cross- Sectional (US)	117 (14 being postmenopausal)	Late pre-, early peri-, late peri-, early postmenopause	STRAW+10	Estradiol FSH	Attention (Digit Span Test), working memory (Letter–number Sequencing Test), verbal fluency (Controlled Oral Word Association), motor function (Grooved Pegboard Test), visuospatial (Hooper Visual Organization Test), and memory (Rey Auditory Verbal Learning Test) Self-reported questionnaires for depression (Beck Depression Inventory II), anxiety (Beck Anxiety Inventory), and overall health (Women’s Health Questionnaire)	Women in the first year of postmenopause performed worse than women in late pre- and late perimenopause on measures of verbal learning, verbal memory, and motor function Women in the first year of postmenopause performed worse than women in late perimenopause on attention tasks Differences in task performances were not explained by menopausal symptoms, mood, or hormone levels
Zhang et al. (2018) [91]	Cross- Sectional (China)	87 (43 being postmenopausal)	Pre- and postmenopause	FSH Menstrual bleeding patterns/history Self-reports of clinical symptoms of menopause	FSH	Alerting, orienting, executive control (Attention Network Task), executive function (Stroop Test), memory (One-Back Working Memory) Resting-state fMRI Anxiety (Self-Rating Anxiety Scale), depression (Beck Depression Inventory-II), sleep (Pittsburgh Sleep Quality Index), and clinical menopausal symptoms (Kupperman Index)	Postmenopausal women had diminished cognitive function as evidenced by longer alerting and executive control times, working memory reaction times, and executive function reaction times than premenopausal women Serum FSH level was negatively correlated with the working memory accuracy Enhanced functional connectivity strength between the left amygdala and bilateral frontal cortex was positively correlated with the executive function accuracy Working memory reaction times were negatively correlated with the decreased functional connectivity strength between the left middle occipital gyrus and the left inferior parietal gyrus

DHEA: Dehydroepiandrosterone; DLPFC: Dorsolateral Prefrontal Cortex; fMRI: Functional Magnetic Resonance Imaging; FSH: Follicle-Stimulating Hormone; IQ: Intelligence Quotient; LH: Luteinizing Hormone; MRI: Magnetic Resonance Imaging; rsDMN: Resting-State Default Mode Network; SHBG: Sex Hormone Binding Globulin; STRAW: Stages of Reproductive Aging Workshop; SWAN: Study of Women’s Health Across the Nation; US: United States. ** Assessed using study-specific tasks.*

A substantial body of literature suggests that objective cognitive decline observed in peri- and postmenopausal women may be associated with menopausal signs and symptoms affecting QOL. The importance of monitoring these variables is demonstrated by the cross-sectional and longitudinal studies listed in Table 4. Overall, menopausal signs and symptoms affect cognitive function in peri- and/or postmenopausal women, but studies are yielding inconsistent results regarding their degree of impact.

Sleep disturbances, stress, anxiety, depression, mood symptoms, and vasomotor symptoms were frequently evaluated to assess their impact on cognitive function in peri- and/or postmenopausal women. The severity of insomnia and sleep disturbances was negatively correlated with cognitive function in peri- [94,95,96,97] and postmenopausal women [95,98]. Memory was especially influenced by stress, measured using the serum cortisol concentration or subjective measures of psychosomatic symptoms, in peri- [99] and postmenopausal [99,100] women. Both subjective and objective measures of anxiety, depression, and mood symptoms were associated with poor cognitive function in peri- [94,101] and postmenopausal [100] women [72,102]. Vasomotor symptoms such as hot flashes influenced cognitive function in peri- [94,96] and postmenopausal [100] women [102]. Lastly, QOL [103] and subjective cognitive complaints [104,105,106] were associated with cognitive function, but the population affected by those factors is unclear due to inconsistent results (Table 4). Future studies should appropriately measure and methodologically report the severity of menopausal signs and symptoms to optimally investigate the correlation between cognitive functions in all stages of menopause.

Lastly, evidence suggests that longer lifetime exposure to estrogen from a longer reproductive life and older age at menopause may attenuate the severity of cognitive decline in women during menopause (Table 5). Overall, better cognitive function in postmenopausal women is associated with a younger age at menarche [107], later age at natural menopause [108], and longer duration of reproductive years [109], but more studies are needed to confirm these findings due to inconsistent results [110]. In addition, the impact of reproductive history on cognitive function in perimenopausal women should be investigated.

**Table 4 nutrients-17-01762-t004:** Studies examining the relationship between menopausal signs/symptoms and objective cognitive measures in peri- and postmenopausal women.

Author	Study Design	Sample Size	Menopause Stage Investigated	Menopause Stage Categorization Criteria	Menopausal Signs/Symptoms	Cognition-Related Measure(s)	Findings
Bojar et al. (2020) [95]	Cross- Sectional (Poland)	300 (143 being perimenopausal)	Peri- and postmenopause	STRAW+10	Insomnia (Athens Insomnia Scale)	Complex memory, verbal memory, visual memory, psychomotor speed, reaction time, complex attention, cognitive flexibility, processing speed, executive function, simple attention, and motor speed (CNS Vital Signs)	Postmenopausal women had a significantly lower reaction time than perimenopausal women The severity of insomnia was negatively correlated with complex and visual memories, and simple attention in peri- and postmenopausal women The severity of insomnia was negatively correlated with simple attention in perimenopausal women
Greendale et al. (2010) [72]	Longitudinal (US)	1903 (59.39% being early perimenopausal)	Pre, early peri-, late peri-, and postmenopause	Menstrual bleeding patterns/history	Depression (Centre for Epidemiologic Studies Depression) Anxiety and vasomotor symptoms (Study Specific Questions) Sleep (Pittsburgh Sleep Quality Index)	Processing speed (Symbol Digit Modalities Test), verbal episodic memory (East Boston Memory Test), and working memory (Digit Span Backward)	All women with high-level depressive symptoms scored one point lower on the Symbol Digit Modalities Test
Grummisch et al. (2023) [96]	Cross- Sectional (Canada)	43	Perimenopause	STRAW+10	Self-reported global cognitive difficulties Forgetting (Memory Functioning Questionnaire) Depression (Centre for Epidemiologic Studies Depression) Stress (Perceived Stress Scale) Sleep (Pittsburgh Sleep Quality Index)	Attention, memory, overall cognitive function (Repeatable Battery for the Assessment of Neuropsychological Status), attention, executive function (Trail-making Test), effort (Test of Memory Malingering), and verbal IQ (National Adult Reading Test)	Subjective measures of attention and memory were negatively associated with baseline perceived stress, mean depressive scores, sleep quality, and number of vasomotor symptoms The number of vasomotor symptoms and sleep quality were negatively correlated with the Trail-making Test outcome
Jaff et al. (2020) [102]	Cross- Sectional (South Africa)	702 (121 being perimenopausal and 277 being postmenopausal)	Late pre-, early peri-, late peri-, early post-, late postmenopause	STRAW+10	Exhaustion, anxiety, irritability, mood, sleep disturbance, and hot flushes (MRS)	Processing speed and incidental recall (Symbol Digit Modalities Test)	Menopausal stages are not associated with processing speed and incidental recall outcomes in sub-Saharan African women Hot flushes and anxiety were associated with a slower processing speed in sub-Saharan African women Severe mood symptoms were associated with a worse performance in incidental recall in sub-Saharan African women
Kalleinen et al. (2008) [98]	Cross- Sectional (Finland)	61 (29 being postmenopausal)	Pre- and postmenopause	Menstrual bleeding patterns/history FSH	Subjective sleep quality, sleepiness, and mood (Study Specific Questionnaire, Basic Nordic Sleep Questionnaire) Objective sleep measures (Spectral Analysis) QOL (EuroQol QOL Questionnaire)	Attention/vigilance (CogniSpeed)	Objective measures of sleep did not differ between pre- and postmenopausal women More REM sleep or earlier REM onset was associated with better CogniSpeed performance in postmenopausal women
Raczkiewicz et al. (2024) [103]	Cross- Sectional (Poland)	287 (141 being perimenopausal)	Peri- and postmenopause	STRAW+10	Overall QOL, general health, physical health, psychological health, social relationships, and environment (World Health Organization QOL Questionnaire)	Complex memory, verbal memory, visual memory, psychomotor speed, reaction time, complex attention, cognitive flexibility, processing speed, executive function, simple attention, and motor speed (CNS Vital Signs) Neurocognitive Index	Postmenopausal women reported having lower/worse overall QOL, physical health, and social relationships than perimenopausal women In perimenopausal women, QOL was positively correlated with visual memory and negatively correlated with processing speed In postmenopausal women, the environment was positively correlated with the Neurocognitive Index and reaction time
Raczkiewicz et al. (2017) [99]	Cross- Sectional (Poland)	300 (143 being perimenopausal)	Early peri-, late peri-, and postmenopause	Menstrual bleeding patterns/history FSH	Stress (serum cortisol concentration)	Complex memory, verbal memory, visual memory, psychomotor speed, reaction time, complex attention, cognitive flexibility, processing speed, executive function, simple attention, and motor speed (CNS Vital Signs)	Serum cortisol concentration did not significantly differ between early peri-, late peri-, and postmenopausal women In postmenopausal women, serum cortisol concentration was negatively correlated with motor speed, psychomotor speed, and reaction time In early perimenopausal women, serum cortisol concentration was positively correlated with processing speed and complex memory In late perimenopausal women, serum cortisol concentration was positively correlated with processing speed
Schaafsma et al. (2010) [104]	Cross- Sectional (Australia)	120 (48 being perimenopausal and 38 being postmenopausal)	Pre-, peri-, and postmenopause	STRAW	Subjective memory and attention complaints and vasomotor symptoms (Study Specific Questionnaire) Vasomotor and menopausal symptoms (Menopausal Symptom Scale) Psychological symptoms (Brief Symptom Inventory, Depression, Anxiety, Stress Scale, and Coping Inventory for Stressful Situations)	Verbal memory (Wechsler Memory Scale), verbal fluency (Controlled Oral Word Association Test), verbal IQ (National Adult Reading Test), visual memory (Benton Visual Performance Test), visuomotor performance (Digit Symbol Coding Test), and attention (CalCAP Computerized Reaction Time Tests)	The menopause stages were not related to the incidence of memory problems Objective verbal memory measures were significantly associated with interference of attention problems and memory problems, and severity CalCAP reaction time measures were significantly associated with attention problems’ interference, duration of memory and attention problems, and spontaneous reports of cognitive problems The objective cognitive measures with significant associations with subjective cognitive complaints included immediate and delayed verbal memory and change in verbal memory between immediate and delayed testing, mean simple reaction time, choice reaction time variability and accuracy, lexical discrimination reaction time accuracy, and selective attention variability
Triantafyllou et al. (2016) [100]	Cross- Sectional (Greece)	39	Postmenopause	Menstrual bleeding patterns/history Estradiol FSH	Psychological, vasomotor, psychosomatic, and sexual symptoms (GCS)	Subjective memory complaint-related question asked during neuropsychological assessment General mental status (MMSE and Clock Drawing Test), verbal episodic memory (Hopkins Verbal Learning Test), and visuospatial episodic memory (Brief Visuospatial Memory Test)	The intensity of psychological and psychosomatic symptoms was negatively correlated with visuospatial episodic memory performance The intensity of vasomotor symptoms was negatively correlated with verbal episodic memory performance The intensity of combined symptoms was negatively correlated with general mental status
Unkenstein et al. (2016) [106]	Cross- Sectional (Australia)	130 (54 being perimenopausal and 40 being postmenopausal)	Pre-, peri-, and postmenopause	STRAW	Attitude to menopause (Menopause Attitude Scale) Depression, anxiety, and vasomotor symptoms (Menopause-Specific QOL Questionnaire) Sleep symptoms (Pittsburgh Sleep Quality Index)	Subjective measurements of memory (Multifactorial Memory Questionnaire and Memory Controllability Inventory) Memory and executive function (Memory Modification of the Boston Naming Test)	Perimenopausal women had a significantly higher frequency of forgetting and less contentment with their memory than pre- and postmenopausal women During perimenopause, women with negative attitude towards menopause and intense depressive, anxiety, vasomotor, and sleep symptoms are more likely to be less content with their memory
Weber et al. (2021) [94]	Longitudinal (US)	85	Early peri-, late peri-, and early postmenopause	STRAW+10 Menstrual bleeding patterns/history	Depression (Beck Depression Inventory-II) Anxiety (Beck Anxiety Inventory) Vasomotor and sleep symptoms (Women’s Health Questionnaire)	Attention (Digit Span Subtest of the WMS-III), vigilance/complex attention (D2 Test of Attention), working memory (Letter–number Sequencing Subtest of the WMS-III), verbal fluency (Controlled Oral Word Association Test), fine motor speed (Grooved Pegboard Test), visuospatial skill (Hooper Visual Organization Test), learning and memory (Rey Auditory Verbal Learning Test)	Four profiles of cognitive function were found, which include normal cognitive function, weakness in verbal learning and memory, strength in verbal memory, and strength in attention and executive function During perimenopause, strength in verbal learning and memory is associated with fewer depressive and vasomotor symptoms compared to normal cognitive function During perimenopause, weakness in verbal learning and memory was associated with more sleep symptoms compared to normal cognitive function
Weber et al. (2012) [105]	Cross- Sectional (US)	75	Perimenopause	Menstrual bleeding patterns/history	Subjective memory function (Memory Functioning Questionnaire) Depression (Beck Depression Inventory-II) Anxiety (Beck Anxiety Inventory) Overall health (Women’s Health Questionnaire)	Attention (Digit Span Subtest of the WMS-III), working memory (Letter–number Sequencing Subtest of the WMS-III), verbal fluency (Controlled Oral Word Association Test), fine motor speed (Grooved Pegboard Test), visuospatial skill (Hooper Visual Organization Test), learning and memory (Rey Auditory Verbal Learning Test)	Subjective memory complaints were associated with greater depressive and anxiety symptoms, more somatic complaints, severe sleep and sexual problems, and decreased feelings of attractiveness Poor performance on the working memory task was associated with frequency of forgetting, seriousness of forgetting, and overall subjective memory complaints Poor performance on the complex attention/vigilance task was associated with frequency of forgetting
Weber et al. (2009) [101]	Cross- Sectional (US)	24	Perimenopause	STRAW	Depression (Beck Depression Inventory-II) Anxiety (Beck Anxiety Inventory) Memory function (Memory Functioning Questionnaire) QOL (Menopause-Specific QOL Questionnaire) Overall health (Women’s Health Questionnaire)	Attention (Digit Span Subtest of the WMS-III), working memory (Letter–number Sequencing Subtest of the WMS-III), verbal fluency (Controlled Oral Word Association Test), fine motor speed (Grooved Pegboard Test), visuospatial skill (Hooper Visual Organization Test), learning and memory (Rey Auditory Verbal Learning Test)	Subjective memory complaints were best predicted by depressive symptoms Objective memory performance was best predicted by depressive symptoms and estrogen level Women with significant memory complaints performed worse on tests of attention, working memory, encoding, and verbal fluency
Yu et al. (2024) [97]	Cross- Sectional (China)	76	Perimenopause	STRAW+10	Insomnia (International Classification of Sleep Disorders)	Sensory processing and attention (Event-Related Potentials)	Women with perimenopausal insomnia had significantly slower response times when working to detect and discriminate targets than controls The degree of neural effort in motor-response initiation and latency time in motor-response execution of the lateralized readiness potential were positively correlated with the severity of insomnia symptoms

CNS: Computerized Neurocognitive Assessment Software; FSH: Follicle-Stimulating Hormone; GCS: Greene Climacteric Scale; MMSE: Mini-Mental State Examination; MRS: Menopause Rating Scale; QOL: Quality of Life; STRAW: Stages of Reproductive Aging Workshop; US: United States; WMS-III: Wechsler Memory Scales-III.

**Table 5 nutrients-17-01762-t005:** Studies examining the relationship between the duration of reproductive life and cognitive measures in peri- and postmenopausal women.

Author	Study Design	Sample Size	Menopause Stage Investigated	Menopause Stage Categorization Criteria	Reproductive History-Related Measures	Cognition-Related Measure(s)	Findings
Gholizadeh et al. (2018) [110]	Cross-Sectional (Iran)	209	Postmenopause	Menstrual bleeding history FSH	Age at menarche Age at menopause	Visuospatial ability, short memory, attention, language, executive function, and orientation (Montreal Cognitive Assessment)	Overall cognitive function in postmenopausal women was not associated with age at menarche and menopause
Karim et al. (2016) [107]	Cross-Sectional (US)	830	Postmenopause	Menstrual bleeding history Estradiol	Age at menarche Date and age at last menstrual period Number of reproductive years	Composite scores for verbal episodic memory (California Verbal Learning Test and East Boston Memory Test), executive function (Symbol Digit Modalities Test, Trail-Making Test, Shipley Abstraction scale, Letter–number Sequencing, and Category Fluency), and overall cognitive performance (Global Cognition)	Age at menarche was negatively associate with global cognition score
Kuh et al. (2018) [108]	Population-based (UK)	1315	Postmenopause	Menstrual bleeding history	Age at period cessation (in months) Age at natural menopause	Verbal memory (Word Learning Task) and processing speed (Visual Search Task)	Verbal memory was positively associated with age at natural menopause
Tierney et al. (2013) [109]	Cross-Sectional (Canada)	126	Postmenopause	Does not specify	Number of reproductive years Age at menarche Age at menopause Years since menopause	Immediate and short delay verbal memory (California Verbal Learning Test), immediate and delayed visual memory (Rey Complex Figure Test), working memory (Digit Span Subtest of the WMS-III), global cognitive function (MMSE), and MCI (California Verbal Learning Test)	Number of reproductive years was positively associated with delayed visual memory, immediate and delayed verbal memory, and working memory performance

MCI: Mild Cognitive Impairment; MMSE: Mini-Mental State Examination; RCT: Randomized Controlled Trial; UK: United Kingdom; US: United States; WMS-III: Wechsler Memory Scales-III.

Our understanding of the degree of change in cognitive performance observed throughout menopausal stages is hampered by methodological limitations in the studies that have been conducted thus far. Future studies may benefit from several considerations. First, the method used to classify the menopausal stage must be consistent across studies, preferably applying uniform criteria. Currently, the recommended gold standard is the application of the Stages of Reproductive Aging Workshop (STRAW)+10 criteria, which were developed as a more comprehensive basis for the assessment of reproductive aging than the original STRAW criteria published in 2001 [11]. Of the 44 reviewed studies, 17 studies used the STRAW+10 criteria, and 4 studies used the STRAW criteria to classify the menopausal stage. In contrast, 3 studies did not specify the criteria used to classify the menopausal stage and 9 studies depended on self-reported menstrual bleeding patterns/history. While bleeding patterns are relevant, there is no gold standard for individuals or clinicians to accurately report menstrual bleeding patterns/history in a way that is comparable across studies. In addition, recall bias is inevitable when utilizing self-reported data only to determine menstrual bleeding patterns/history. Due to challenges with measuring the exact volume of blood loss and accurately reporting the timing of changes in bleeding patterns, future studies must determine better methods to measure and report menstrual bleeding patterns and utilize the STRAW+10 criteria.

Second, future studies must consider utilizing different approaches to measure cognitive function in this population, especially to confirm subjective experiences of changes in cognitive function and affect. Of 44 studies, 3 studies utilized interviews or study-specific surveys only to test their hypotheses. While qualitative data from interviews and study-specific surveys provide valuable insights, there is a need to further refine and confirm those findings by triangulating evidence with different approaches [111]. For instance, future studies may consider supplementing qualitative data with data obtained via validated questionnaires conducted by clinicians and/or researchers and/or computerized cognitive tasks.

Third, it is possible that the discrepancies observed in different studies may be partly explained by the heterogenous nature of menopause itself. Certain changes in the central neuroendocrine system and ovaries are specific to each stage of menopause, which may explain why different cognitive domains are affected in different stages of menopause [43,83,112]. Evidence suggests that trajectories of estradiol and FSH vary across pre-, peri-, and postmenopausal women, which may partially explain the varying degrees of signs and symptoms observed in postmenopausal women [16]. Yet, it is unclear how changes in cognitive function and affect may interact with the heterogenous nature of menopause itself. Thus, longitudinal studies identifying the degree of biological heterogeneity throughout the course of menopause would help to build a strong foundation for future studies investigating their impact on women’s cognitive function during such a critical period. Lastly, relations between hormone levels and cognitive outcomes were assessed using different sets of hormones and cognitive tests across different studies, which introduces a challenge when trying to interpret and compare results. Also, in addition to hormones affecting changes in cognitive function in menopausal women, there is a need to identify other contributing factors of cognitive decline in this population.

## 5. Discussion and Implications for Future Studies

Current evidence indicates that the peri- and postmenopausal stages are accompanied by changes in women’s cognitive function. Due in part to the disparate experiences of menopausal signs and symptoms influenced by hormonal fluctuation, women during the menopausal stages are at risk of cognitive decline. For perimenopausal women with heavy bleeding, the risk of cognitive decline may be exacerbated with heavy blood loss due to ID/IDA, but more studies are needed to understand the role of iron status in cognitive decline for menopausal women. Evidence suggests that ID is related to cognitive changes in young women and that iron supplementation can reverse these cognitive changes. Whether or not iron supplementation during the peri- and postmenopausal stages can alleviate the cognitive decline experienced at these timepoints is a question that has yet to be answered, especially in those with ID. To ensure that future studies are optimally informative, it will be critical for investigators to appropriately measure and methodologically report sociodemographic variables and to choose and interpret cognitive tests appropriately for the target population.

Because cognitive decline during menopause has been understudied, the published literature lacks detailed descriptions of the diverse cognitive experiences of people undergoing menopause. Future studies should consider a multitude of factors that could contribute to cognitive decline during the menopause transition. Factors such as lower educational attainment, untreated mental health disorders, high trauma exposure, substance use, lack of access to healthcare, infectious diseases [113], and adverse childhood experiences [114] contribute to cognitive decline, and women with these experiences may be disproportionately affected by menopause-related cognitive decline. Understanding the additional sociodemographic and mental health-related cognitive decline components, of which few studies have measured, is essential for ensuring the optimal treatment of cognitive decline during the menopause transition.

Choosing and interpreting cognitive tests that are appropriate for diverse target populations is challenging. While most research has shifted away from the original prejudicial creation of cognitive tests, biases still abound in cognitive tests and their interpretation [115]. Basic science and research in cognitive psychology have begun to consider equity and fairness, but the translation of equitable processes to cognitive assessment research and practice is slow [116]. Future studies must include diverse populations and use cognitive tests that will accurately measure cognition in the target population so that the interpretation of the findings is not clouded by test biases. Finally, future studies that include diverse populations must also consider the role of discrimination and racism in cognitive decline [117].

Lastly, the accurate interpretation and comparison of study findings would be facilitated by applying standardized definitions for categorizing women as being in the pre-, peri-, or postmenopausal stages. Current studies have not used a standardized way of classifying women, although most studies have utilized the Stages of Reproductive Aging Workshop+10 criteria [11] along with menstrual bleeding patterns or history and/or a set of endocrine measures. While each criterion serves its purpose, applying a different combination of criteria to categorize women’s menopausal stages may be contributing to inconsistent findings across studies.

Iron supplementation holds promise for alleviating symptoms occurring during the peri- and postmenopausal stages of a woman’s life, including cognitive declines. This relatively cheap and simple intervention may offer a significant way to help women maintain functional ability as they age. This will not only improve the quality of many women’s lives but could also have a positive effect on the economy by reducing the number of missed workdays related to menopause. For these reasons, it is imperative that future studies are conducted to fill the gaps in our understanding. Only then will we be able to determine if iron supplements should be considered as a treatment during the menopause transition.

## Figures and Tables

**Figure 1 nutrients-17-01762-f001:**
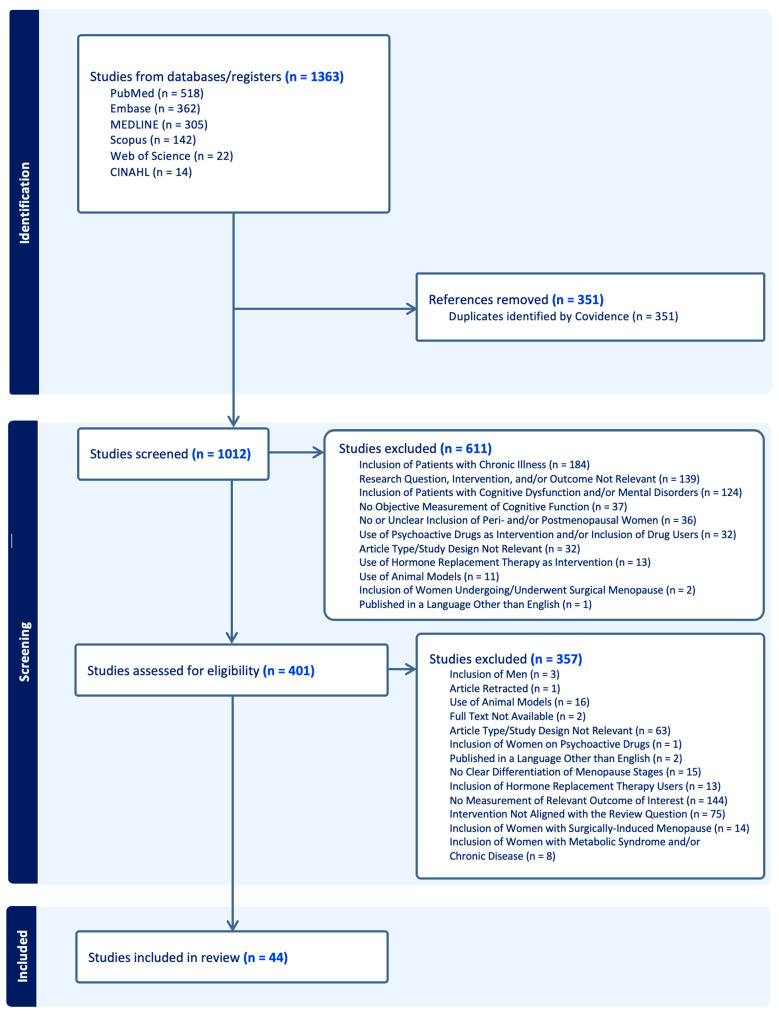
PRISMA flowchart.

**Table 1 nutrients-17-01762-t001:** Studies examining the relationship between iron status and cognitive function in perimenopausal women.

Author	Study Design	Sample Size	Menopause Stage Investigated	Menopause Stage Categorization Criteria	Iron Biomarkers	Cognition-Related Measure(s)	Findings
Valentina et al. (2013) [49]	Observational follow-up of double-blind, placebo- controlled RCT (France)	3932 (1431 being pre- and perimenopausal)	Pre-, peri-, and postmenopause	Does not specify	Ferritin Hemoglobin	Episodic memory (RI-48 Cued Recall Test), semantic memory (Phonemic & Semantic Fluency Task), working memory (Backward & Forward Digit Span), and mental flexibility (Delis–Kaplan Trail-Making Test)	In pre- and perimenopausal women aged 46 and older at baseline, there was a significant inverse association between ferritin and overall cognitive measures as well as working memory There was an inverse association between ferritin and phonemic fluency in postmenopausal women
Barnett et al. (2025) [50]	Cross- sectional * (US)	27 (8 being early perimenopausal)	Perimenopause	STRAW+10 FSH	Ferritin Hemoglobin Hematocrit Mean Corpuscular Volume Mean Corpuscular Hemoglobin Red Cell Blood Count Red Cell Distribution Width	Immediate and delayed recall (Face/Name Associative Memory Task), learning (Probabilistic Selection Task), memory (Rule-Based Category Learning Task), and working memory (Visuospatial Working Memory Task) Concurrent and resting electroencephalographic measures Structural MRI	A higher serum ferritin percentile was associated with higher accuracy, higher discriminability, and shorter reaction times in all cognitive tasks in perimenopausal women Higher values of iron biomarkers associated with oxygen transport were related to better cognitive performance in perimenopausal women

FSH: Follicle-Stimulating Hormone; RCT: Randomized Controlled Trial; STRAW: Stages of Reproductive Aging Workshop; US: United States. ** Originally, this study was designed using a factorial design based on participants’ iron statuses and menopausal statuses. Due to challenges with recruitment, however, the dataset published is suitable for exploring potential correlations, not testing the original hypotheses.*

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
