# Peer review of "Cognitive Function in Peri- and Postmenopausal Women: Implications for Considering Iron Supplementation"

_nutrients, 2025, doi:10.3390/nu17111762_

Round 1

Reviewer 1 Report

Comments and Suggestions for Authors

The manuscript presents an important and timely discussion on the cognitive challenges faced by women during the menopausal transition and the potential role of iron supplementation as an alternative to hormone replacement therapy (HRT). The authors effectively highlight the limitations of current HRT recommendations for cognitive symptoms and point out the physiological basis for considering iron as a possible intervention, particularly in light of blood loss during perimenopause.

However, the review would benefit from greater clarity regarding the selection and evaluation of sources. A detailed graphical representation or flowchart illustrating how studies were chosen, assessed, and synthesized would significantly improve the transparency and rigor of the literature review. Such a figure could help readers better understand the inclusion/exclusion criteria and the overall strength of the evidence.

Additionally, it would be valuable for the authors to explicitly discuss the methodological limitations of the studies referenced, particularly those relating to sample size, study design, population heterogeneity, and outcome measurement. This would provide a more balanced view of the current evidence and further underscore the need for well-designed clinical trials in this area.

Overall, the topic is highly relevant, and the rationale for exploring iron supplementation is compelling. Strengthening the methodological transparency and critically assessing the limitations of existing studies would enhance the manuscript's contribution to the field.

Author Response

We thank the reviewers for their thoughtful comments and have responded to each, below. We have provided two versions of our revised manuscript: one “clean” version and one version where we’ve highlighted all changes (additions as well as anywhere that we reorganized the text). We feel that the changes made as a result of the reviewers’ comments have strengthened the manuscript. Thank you for this opportunity.

Comment 1: The manuscript presents an important and timely discussion on the cognitive challenges faced by women during the menopausal transition and the potential role of iron supplementation as an alternative to hormone replacement therapy (HRT). The authors effectively highlight the limitations of current HRT recommendations for cognitive symptoms and point out the physiological basis for considering iron as a possible intervention, particularly in light of blood loss during perimenopause.

However, the review would benefit from greater clarity regarding the selection and evaluation of sources. A detailed graphical representation or flowchart illustrating how studies were chosen, assessed, and synthesized would significantly improve the transparency and rigor of the literature review. Such a figure could help readers better understand the inclusion/exclusion criteria and the overall strength of the evidence.

Additionally, it would be valuable for the authors to explicitly discuss the methodological limitations of the studies referenced, particularly those relating to sample size, study design, population heterogeneity, and outcome measurement. This would provide a more balanced view of the current evidence and further underscore the need for well-designed clinical trials in this area.

Overall, the topic is highly relevant, and the rationale for exploring iron supplementation is compelling. Strengthening the methodological transparency and critically assessing the limitations of existing studies would enhance the manuscript's contribution to the field.

Response 1: Thank you for this feedback. We value and agree with the reviewer’s input. Therefore, we have created a separate section in our manuscript titled Literature Search after the Introduction to provide clarity regarding the selection and evaluation of sources. This section lists search terms used for the literature search (lines 129-132, pg. 3) and criteria for the literature inclusion and exclusion (lines 132-153, pg. 3-4). Additionally, we added details to and refer to Figure 1 (pg. 5), which is a graphical flowchart illustrating the decision-making process of the literature search. Finally, to strengthen the methodological transparency and critically assess the limitations, per the reviewer’s recommendation, we added columns on sample size and study design to our tables and added lines 342-384 (pg. 36) to discuss the methodological limitations of the referenced studies.

Comment 2: The title of this 31-page manuscript focuses on the relationship between iron status and cognitive function in pre- and post-menopausal women and fits the theme of the special issue.   However, the introduction has just over a page on the role of iron in cognitive function (bottom of page 2 and page 3).  Just over a page describes iron in pre- and post-menopausal periods (pages 23 and 24).  Other than the small last paragraph of the conclusion, the review does not describe iron.  The manuscript does not describe what the title says.  It is a review of cognitive function in pre- and post-menopausal women.  Thus, the manuscript does fit the special issue on iron and brain function. 

Response 2: Thank you for this feedback. After discussion, we decided that a change in our title was warranted and it now reads, “Cognitive Function in Peri- and Postmenopausal Women: Implications for Considering Iron Supplementation”. This decision was based on the fact that the evidence explicitly evaluating iron supplementation in peri- and postmenopausal women is limited, so we changed our title to more precisely describe our review. In addition, we moved the section titled Iron in the Peri- and Postmenopausal Periods, so that it appears earlier in the manuscript (line 156, pg. 6) and created a table (Table 1, pg. 8) summarizing the 2 studies that have tested the impact of iron on women’s cognitive function during the peri- and postmenopausal periods.

Comment 3: The manuscript submitted by Choi et al., titled: "The Association between Iron Status and Cognitive Function in Peri- and Postmenopausal Women: Implications for Future Studies" is a review article aiming to discuss the evidence in the literature in regards to the association between iron status and cognitive function in peri-and post-menopausal women. The article is well written and structured with a good flow and has considered the literature broadly. 

The reviewer would like to bring up a couple of conceptual points that in his professional opinion should be addressed and discussed more in depth in the paper and would thus strengthen the paper significantly. 

One important item in terms of cognition is the definition of. Cognition is a complex ability and is assessed in different ways respective to the dimension(s) that one wishes to focus on or assess specifically. Moreover, symptoms that appear to reduce cognitive and/or perceptive ability may have more than one causes even though the symptoms may be similar and be manifested similarly. Therefore the aspect for example of fogginess or disorientation can be due to several reasons including hormonal deregulation due to the peri/post menopausal stages. It may or may not directly be due to iron status. Also even if iron status is influencing the measured outcome one needs to discuss confounders which may also contribute to such symptomatic manifestation.

An interesting discussion would be on the hormonal replacement and if there is evidence to suggest that symptoms improve with such course of action. Also the level of menses decline needs to be discussed in factoring in the effect of the menopausal condition to iron status. 

Response 3: Thank you for this positive feedback. The reviewer is absolutely correct in saying that cognition is a complex ability and is assessed in different ways respective to the dimension(s) that one wishes to focus on or assess specifically. After reading the reviewer’s comment, we realized that we must not have made this point clearly in the previous manuscript. Therefore, we moved up the definition of cognition to line 76 (pg. 2).

We agree that a discussion regarding the impact of hormone replacement therapy on peri- and postmenopausal women’s cognitive function would be interesting, but we feel such a discussion would be outside the scope of this review and, therefore, have not added it here. That said, we hope future articles and/or studies will capture and describe this relationship in detail.

Lastly, we agree that it would be ideal to document the level of menses decline over the course of menopause. However, the studies we reviewed did not assess the level of menses decline due in part to large variability and difficulty in quantifying (for example, women with several months of amenorrhea may suddenly experience heavy bleeding and vice versa). In addition, a validated method to accurately measure bleeding has not been agreed upon, to our knowledge. As such, we are unable to discuss the level of menses decline when reviewing the literature to date.

Reviewer 2 Report

Comments and Suggestions for Authors

The title of this 31-page manuscript focuses on the relationship between iron status and cognitive function in pre- and post-menopausal women and fits the theme of the special issue.   However, the introduction has just over a page on the role of iron in cognitive function (bottom of page 2 and page 3).  Just over a page describes iron in pre- and post-menopausal periods (pages 23 and 24).  Other than the small last paragraph of the conclusion, the review does not describe iron.  The manuscript does not describe what the title says.  It is a review of cognitive function in pre- and post-menopausal women.  Thus, the manuscript does fit the special issue on iron and brain function. 

Author Response

(The authors gave the same response as above.)

Reviewer 3 Report

Comments and Suggestions for Authors

The manuscript submitted by Choi et al., titled: "The Association between Iron Status and Cognitive Function in Peri- and Postmenopausal Women: Implications for Future Studies" is a review article aiming to discuss the evidence in the literature in regards to the association between iron status and cognitive function in peri-and post-menopausal women. The article is well written and structured with a good flow and has considered the literature broadly. 

The reviewer would like to bring up a couple of conceptual points that in his professional opinion should be addressed and discussed more in depth in the paper and would thus strengthen the paper significantly. 

One important item in terms of cognition is the definition of. Cognition is a complex ability and is assessed in different ways respective to the dimension(s) that one wishes to focus on or assess specifically. Moreover, symptoms that appear to reduce cognitive and/or perceptive ability may have more than one causes even though the symptoms may be similar and be manifested similarly. Therefore the aspect for example of fogginess or disorientation can be due to several reasons including hormonal deregulation due to the peri/post menopausal stages. It may or may not directly be due to iron status. Also even if iron status is influencing the measured outcome one needs to discuss confounders which may also contribute to such symptomatic manifestation.

An interesting discussion would be on the hormonal replacement and if there is evidence to suggest that symptoms improve with such course of action. Also the level of menses decline needs to be discussed in factoring in the effect of the menopausal condition to iron status. 

Author Response

(The authors gave the same response as above.)

Round 2

Reviewer 2 Report

Comments and Suggestions for Authors

The manuscript as I indicated in my first review was except it did not fit the title nor meet the qualifications of the special issue.  With the title change, movement of material, and changes to address the other reviewers' comments.  The manuscript is publishable.  However, I still question the relevance to the special issue.